# Development of a new perceived injustice scale for Bangla speaking population

**Chandrima Mourin, Muhammad Kamruzzaman Mozumder** *

Department of Clinical Psychology, University of Dhaka, Dhaka, Bangladesh

* mozumder@du.ac.bd

## Abstract

### Background

Perceived injustice is a relatively novel psychosocial construct starting to get some attention among researchers studying health and mental health outcomes. In the context of the widespread perception of being a victim of injustice in Bangladesh, a gap in instruments measuring perceived injustice was evident. The novelty of the construct and lack of similar instruments necessitated the development of a new Perceived Injustice Scale for Bangla speaking population.

### Method

A three-stage approach was used to develop the instrument. A thorough review of literature coupled with interviews with the general population as well as professionals contributed to the conceptualization of the construct. A rigorous process of expert evaluation and item analysis resulted in the identification of the most suitable items.

### Results

The scale demonstrated a single-factor structure with adequate evidence of its reliability and validity. Reliability of the scale was assessed using internal consistency method (Cronbach alpha = .931; p< .01) and test-retest method (r = .837; p< .01). While validity was assessed using criterion-related validity (correlation with the Steel Injustice Inventory at r = .428; p< .01) and construct validity (correlation with the Perceived Stress Scale at r = .332; p< .01; and with the WHO Wellbeing Index at r = -.361; p< .01).

### Conclusion

The strong evidence of reliability and validity suggests the suitability of the scale for measuring perceived injustice in Bangladesh. Researchers and clinicians can use this 10-item scale with confidence in assessing perceived injustice among the general population.

**Data Availability Statement:** Complete data used in this manuscript are collected by the authors. All the data have been made public and is available at https://doi.org/10.17605/OSF.IO/4Y7FC.

**Funding:** The author(s) received no specific funding for this work.

**Competing interests:** The authors have declared that no competing interests exist.

## Introduction

Injustice is a universal phenomenon that relates to unfairness or undeserved and is found to be directly linked to a number of physical and mental health outcomes [1]. Although the exact medical, psychological, and social reasons underlying this association are yet unknown, findings suggest that repeated experiences of injustice in the family, work, or in society can lead to unfavorable stress-related reactions that eventually raise the risk of a heart attack [2]. The feeling of injustice varies from person to person due to their perception, making it important to understand and measure perceived injustice.

In the general sense, injustice can be construed as the lack or absence of fairness or justice [3]. In legal terms, injustice generally refers to the absence of justice, unfairness or violation of others' rights. This definition requires the actual presence of unfairness or evidence of the violation of the rights of other people. On the other hand, perceived injustice can be referred to as the perception of getting disproportionate treatment from others by which a person can feel unfairness or disrespect. Older definition of perceived injustice from a medical research perspective described it as "a negative appraisal regarding irreparability and severity of loss associated to pain, and feelings of blame and injustice" [4], More recently, perceived injustice has been defined as "a belief that one has been treated unfairly and disrespectfully and is suffering unnecessarily as a result of another person's action" [5].

Although perceived injustice differs from real injustice, it can create an equally unfavorable psychological effect on a person. When a person perceives him or herself as a victim of injustice, the conviction is likely to produce psychological effects (including emotional, behavioral, and social responses) in the person similar to being an actual victim [6]. Perceived injustice seemingly holds a more significant position as a social phenomenon. However, in recent time it is being discussed and studied as a novel and useful concept in medicine and psychiatry [7]. Research findings demonstrated a link between perceived injustice and mental health outcomes such as symptoms of depression, post-traumatic stress, and social phobia [8, 9]. Moreover, psychotherapeutic interventions (such as Acceptance and Commitment Therapy) are being tested and proven effective in reducing perceived injustice [10].

Perceived injustice is a little-researched area in behavioral science. Therefore, there is a visible scarcity of instruments for measuring perceived injustice in research and clinical contexts. Sullivan, Adams [4] developed the 12-item Injustice Experience Questionnaire (IEQ) to measure the experience of perceived injustice associated with injury. Apart from identifying perceived injustice, the IEQ has been claimed to support decision-making on necessary psychological intervention for the patient [11]. Neumann, Berger [12] developed the Perceived Injustice Questionnaire (PIQ) consisting of four subscales to measure relevance and perception of injustice among people who have experienced violence in war and conflict. The psychometric properties of this scale suggest its suitability for use in psychotherapeutic settings of trauma and conflict [12]. Steel (personal communication, 2024) developed a brief four-item instrument to measure the experience of injustice. This inventory assesses injustice in the personal and occupational domains. Hodson, Creighton [13] developed a single factor four-item Likert-type scale that measures employees' perception of being treated unfairly by the employers. Instead of focusing on a generalized evaluation of injustice, this assessment includes four specific questions for the employees to describe the degree of injustice they are facing in their workplaces.

The tools for measuring perceived injustice as discussed above were developed to fit with specific contexts including chronic pain [11], war and conflict [12], or workplace [13]. None of these seems to be a useful tool for assessing perceived injustice in a general context. Except for a translated version of the Steel Injustice Inventory (Steel, personal communication, 2024),

no other instruments for measuring perceived injustice have been used in Bangladesh. All the instruments described above are developed in exotic contexts, and it is well known that social perception varies from social circumstances and cultural context. Therefore, the development of a homegrown perceived injustice instrument suitable for the Bangladesh context felt necessary. The development of such a tool is expected to increase research interest and activities involving perceived injustice.

## Method

### Research design

A sequential process of tool construction [see 14] was used in developing the perceived injustice scale. This procedure was divided into three main stages, i) understanding of the construct and item generation, ii) item selection, and ii) assessment of psychometric properties (see Fig 1).

### Participants

For determining sample size, the rule of thumb 'N = 50+8m' for survey study was used ('N' is the required sample size and 'm' is the number of variables or items) [see 15]. This calculation suggested an approximate sample size of 226 for the present research. Data from four participants were discarded due to incomplete response. The final sample was comprised of 221 adults (112 male and 109 female) recruited using the purposive sampling technique. Among them, 36 participants were interviewed twice to assess the stability (test-retest) reliability of the instrument.

### Instruments

In the process of assessing the psychometric properties of the proposed Perceived Injustice Scale, a few available instruments were used.

Steel Injustice Inventory (Zachary Steel; personal communication, 2024). The Bangla translated version of the instrument was used to assess the criterion validity of the upcoming perceived injustice instrument. This inventory contains four items assessing injustice in general and in the workplace.

Perceived Stress Scale [PSS-10; 16]. The Bangla version of PSS-10 [17] was used as a measure of perceived stress. The Bangla PSS-10 demonstrates adequate construct validity (with GHQ 28, r = .579, p< .01) and reliability (Cronbach's alpha = .715 and test-retest r = .745, p < .01). It was hypothesized that perceived injustice will result in the feeling of stress and therefore, the scale score will have a high positive correlation with scores on the PSS-10.

WHO Well-Being Index [WHO-5; 18]. The Bangla version of WHO-5 [19] was used to measure well-being. The Bangla WHO-5 has been reported with adequate psychometric properties (Cronbach's alpha = .754; test-retest r = .713, p < .01; convergent validity using Warwick-Edinburgh Mental Well-Being Scale, r = 0.542). It was hypothesized that people with a high perception of injustice will have lower well-being and thus WHO-5 score can be used to assess the convergent validity of the perceived injustice scale.

Composite questionnaire. A composite questionnaire was used to collect data on socio-demographic variables (e.g., age, gender, educational level, occupation, socio-economic status, area of residence, marital status, physical illness, experiences of receiving mental health service, lack of pleasure, lack of interest, and experiences of being a victim of violence) and two single-item generic indicators for assessing experience of injustice.

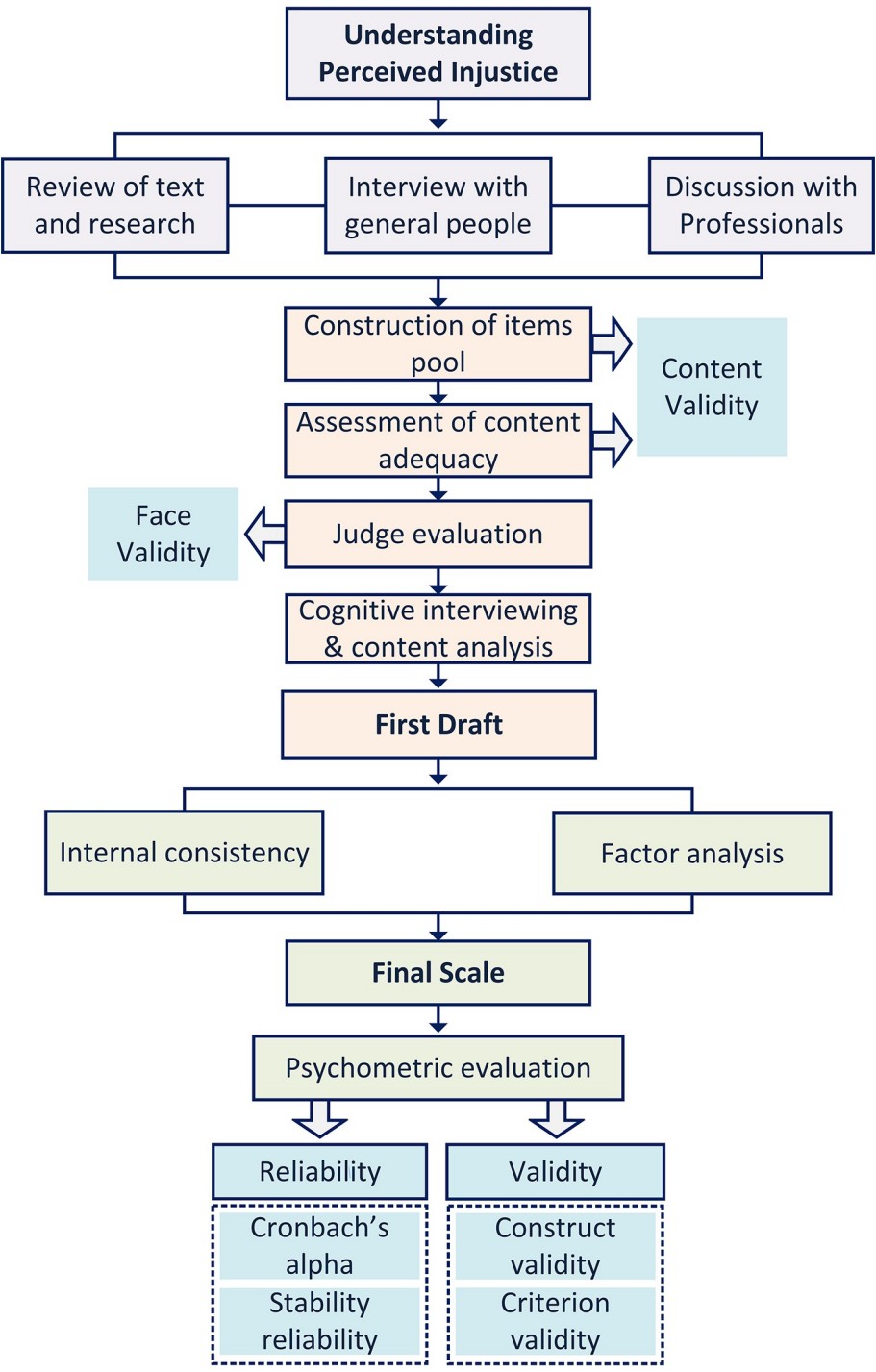

**Fig 1. The multi-stage procedure of scale development.**

## Procedure

Data was collected through both paper-pencil and online (using Google form) format over a period of one and a half months (7 August 2023 to 30 September 2023). Written understood consent was taken from the participants using explanatory statements detailing the purpose,

process, and required activities in the research along with information on the voluntary nature of participation, rights to withdraw, privacy and confidentiality. Refusal and non-completion rates were less than 10%, however, reminders were used to ensure minimal non-completion among the participants. The participants joining through the survey link indicated their consent by accepting to proceed with the study after reading the explanatory statements. This study maintained all the universal principles of human research ethics, which are autonomy, justice, beneficence and non-maleficence. Ethical approval from the Department of Clinical Psychology, University of Dhaka (project number: MS230201 date of approval: April 4, 2023) was received prior to the start of data collection.

## Findings

The process of development and validation included the use of both qualitative and quantitative approaches. PASW 18 [20] software package for statistical analysis was used in analyzing quantitative data. The findings are presented in line with the three distinct stages that were used in developing and validating the perceived injustice inventory.

## Stage I: Understanding the construct and item generation

Limited use of the term perceived injustice in the academic or research context of Bangladesh demanded an exploratory study to identify components and manifestations of perceived injustice before any item can be generated for the instrument. Three converging methods including i) review of text and research, ii) interview with general people, and iii) discussion with social science and mental health professionals, were used to gain a clear understanding of the concept and experience of injustice in Bangladesh context.

The exploratory process led to the drafting of 62 initial items which were reviewed by 16 trainee clinical psychologists for personal reflection on the items. The items were then divided into cognitive, emotional and behavioral domains for better conceptualization. After careful revisions and content adequacy assessment, the initially constructed 62 items were reduced to 50.

## Stage II: Item selection

The selection of items for the final instrument from the items pool was done through a dual process involving judge evaluation and item analysis.

## Judge evaluation

The 50 items were presented to 16 judges with expertise in social science and mental health to assess the face validity for evaluating the appropriateness of the items in measuring perceived injustice. In selecting the judges, a wide range of variation was considered. The expert panel consisted of experts from social science and mental health background with the majority of them working in the academic field. Among the 16 judges, two (12.5%) were from psychiatry, four (25%) from social science, and the remaining ten (62.5%) were from the clinical psychology field. The average scores (on a scale of 1–4) given by the judges were calculated for each item. An average score > 2 was used as the cutoff value for the selection of the items. Twenty-six items passed the selection criterion and subsequent analysis of the comments from judges about the items were carried out to achieve further precision of the instrument. One item (with an average rating of 2.33) was removed as per judges' comments on its lack of context specificity while another (with an average rating of 2.06) was removed for comments on its lack of representativeness. Cognitive interviewing followed by content analysis of the items

resulted in the merger and reduction of nine items into four. This process resulted in a revised version i.e., the first draft of the scale with 19 items.

## Item analysis

Item analysis was carried out to evaluate the quality of the selected items. The process involved inter-item correlation, corrected item-total correlation, and exploratory factor analysis.

*Inter-item correlations*. Inter-item correlation or item-item correlation was checked to determine the association between the items. Acceptable levels of inter-item correlations (r = .391 to .848) among the items indicated a moderate to high level of association between the items. High correlations (r $\geq$ .8) indicate concerns of redundancy of items. Additionally, an initial assessment of internal consistency suggest that the 19-item scale would have an overall Cronbach's alpha of .971 providing further indication of overly similar items. Therefore, to attain a parsimonious scale, further scrutiny of inter-item correlation matrix was carried out. Correlation coefficient was rounded to single decimal point and multiple items were found with overly high correlation. Item # 10 demonstrates high correlations (r $\geq$ .8) with seven others items (4, 5, 6, 7, 8, 11, 12) which also had high inter-correlation within themselves. Item # 16 shows similar high correlation with two other items (15 & 17). It was obvious that the two key items (10 & 16) would be able to represent the remaining and hence the nine items (4, 5, 6, 7, 8, 11, 12, 15 & 17) was removed.

*Corrected item-total correlations*. Corrected item-total correlations for the ten items were calculated to identify the level to which each item contributes to the overall construct (i.e., perceived injustice) that is being measured. All the ten items were found to have sizable correlations ranging from .543 to .841 (all significant at p < .01) between the items' score and the total score of all other items on the scale.

*Factor analysis*. Exploratory Factor Analysis was conducted to identify the latent patterns in data and to reduce the number of items. The suitability of the data set for exploratory factor analysis (EFA) was demonstrated by Kaiser-Meyer-Olkin (KMO) measure of sampling adequacy at .946 which was above the recommended value of 0.6. Bartlett's test of sphericity also indicated the suitability of factor analysis. The shared variance was indicated by commonalities where the values of all items were above 0.3. It also indicated that the 10-item scale can explain 61.95% of total variance. Principle component analysis method was used in EFA. Inspection of eigenvalues, scree plots and parallel analyses suggested a single-factor solution for the scale. No removal of items was suggested in factor analysis, and therefore, the ten items comprised the final version of the new perceived injustice scale.

## Stage III: Assessment of psychometric properties

Reliability and validity are the two key psychometric properties of a tool that were assessed for the newly developed instrument.

## Validity of the perceived injustice scale

Content, criterion, and construct validity were assessed for the newly developed instrument. Content validity was ensured by the exploratory component used in generating the items. Additionally, the use of an expert panel from multiple disciplines who checked face validity indirectly supports the content validity of the instrument.

For measuring criterion validity, the translated version of Steel Injustice Inventory was used. A good correlation (r = .428; p < .01) between the total scores on the new instrument and the Steel injustice inventory supports criterion related validity for a newly developed perceived injustice inventory. A moderate positive point-biserial correlation was found (r = .426;

p < .01) between the scores on the perceived injustice scale and experience of injustice (dichotomous response option—yes or no). This suggests that the participants who have experienced injustice scored higher on the perceived injustice scale. Correlation with overall rating (0–10) of the feeling of injustice and score on the perceived injustice scale also indicates a moderate positive relation (r = .441; p < .01). These correlations provide additional support for the criterion validity of the instrument.

Among the two types of construct validity, the convergent method was used. Sufficient correlation with measures of perceived stress (r = .332, p < .01) and well-being (r = -.361, p < .01) with the perceived injustice scale provide evidence of construct validity of the newly developed instrument.

## Reliability of the perceived injustice scale

Internal consistency and stability (using the test-retest method) reliability were assessed for the instrument. Cronbach's alpha was used as the indicator of internal consistency among the ten items [12]. Excellent internal consistency reliability of the scale was demonstrated with Cronbach's alpha at .931. Two consecutive administrations with an approximate gap of three weeks on a sub-sample of 36 respondents were conducted to assess the stability reliability of the scale in the test-retest method. Excellent stability reliability was demonstrated through high correlation (r = .837; p < .01) between the test and re-test scores of the participants [21].

## Discussion

In the context of a growing and widespread presence of perception of injustice in the Bangladeshi population and the worldwide limited availability of tools to measure perceived injustice, a need for developing a scale was felt. The novelty of the construct required an exploratory component in generating items. The ideas generated from the review of literature, mind-map exercises in the research team, and informal interviews with professionals as well as the general population contributed to the development of the initial pool of items. This variability of sources ensured the comprehensiveness of the items and reduced the probability of overlooking important aspects [22]. The inclusion of multiple sources of data ensured the content validity of the scale. During the expert evaluation, the judges checked the face validity which further supplemented (indirect) assurance of content validity of the items for measuring perceived injustice. A rigorous two-step item selection process ensured the inclusion of items with high relevance. This was further demonstrated by high inter-item correlation for all the items (both initial and final) during the item analysis process. Exploratory factor analysis indicated a single-factor structure of the instrument, suggesting that the items measure only one construct which is 'perceived injustice'.

The newly developed 10-item scale was administered along with the Bangla translated Steel injustice inventory. A significant correlation (r = .428; p< .01) between the scores of the two instruments provided evidence for criterion related validity of the scale. This scale showed a positive correlation with perceived stress (r = .332; p < .01) and a negative correlation with well-being (r = —.361; p< .01). These significant correlations of perceived injustice scale score, in the pre-hypothesized direction, with the two constructs, perceived stress and wellbeing support the scale's construct validity in the convergent method.

The scale demonstrated stability of assessment across different administrations over time (correlation of score between test and retest was r = .837; p < .01) which indicates excellent stability reliability of the scale [21]. The scale also demonstrated excellent internal consistency reliability (Cronbach's alpha = .931) [23]. This score can be compared to other established injustice scales. For example, the overall Cronbach's alpha for the Perceived Injustice

Questionnaire (PIQ) was .929 [12], for the Hodson et al.,'s perceived injustice tool was .70 [13], and the test-retest reliability for the Injustice Experience Questionnaire (IEQ) was .90 [4].

The psychometric property of the newly developed perceived injustice inventory was found to be superior to other instruments namely the PIQ and the Hodson et al.,'s tool in measuring perceived injustice. The scale demonstrated sufficient evidence of validity and reliability comparable to the other widely used locally developed instruments such as the Anxiety Scale [24], Zahir Depression Scale [22], and the Dhaka University Obsessive Compulsive Scale [14]. The perceived injustice inventory can therefore be assumed to gain acceptability among the researchers and clinicians working on understanding and assessing perceived injustice.

This article presents the process of development of a contextually relevant new instrument for measuring perceived injustice. It also presented evidence of the instrument's psychometric properties claiming its position as a reliable and valid instrument. Though beyond the scope of a single study, the tool will need to gather further evidence of its construct validity (using a multi-trait muti-method approach) and generalizability (i.e., testing for its property across different population groups). Studies directed towards that goal may further enhance the utility and credibility of the instrument. Building up of evidence may lead to further enhancement of the instrument (through revision). With increasing recognition of perceived injustice as a relevant phenomenon in mental health, psychiatry and medicine [4, 7–10], the newly developed tool is expected to be welcomed among the researchers and clinicians from these field of study. It would be useful for clinical and research purposes if normative data could be collected and cutoff scores could be established for screening or assessing the severity of perceived injustice.

### Constraints on generality

The scale was developed based on data collected from a sample of adult participants using purposive sampling. The lack of systematic sampling challenges the representativeness of the data which was needed for generalizability or robustness of the instrument. Therefore, the findings may not generalize to the broader population.

### Conclusion

To our knowledge, it is the first tool for measuring perceived injustice in Bangladesh. Comprised of 10-item that are phrased with everyday language, the scale is convenient for self-administration requiring 5–6 minutes to respond to all the items. Excellent psychometric properties demonstrated by this newly developed scale are likely to ensure its confident use as a self-report assessment tool. The development of this contextually relevant perceived injustice scale contributes to the indigenization of psychology in Bangladesh. The availability of this tool created possibilities for future research and theorizing on this relatively novel psychosocial construct.

### Supporting information

**S1 File. Mourin perceived injustice scale.**
(PDF)

**S2 File. Mourin perceived injustice scale (English translation).**
(PDF)

### Author Contributions

**Conceptualization:** Muhammad Kamruzzaman Mozumder.

**Data curation:** Chandrima Mourin.

**Formal analysis:** Muhammad Kamruzzaman Mozumder.

**Methodology:** Chandrima Mourin, Muhammad Kamruzzaman Mozumder.

**Resources:** Chandrima Mourin.

**Supervision:** Muhammad Kamruzzaman Mozumder.

**Visualization:** Muhammad Kamruzzaman Mozumder.

**Writing – original draft:** Chandrima Mourin.

**Writing – review & editing:** Chandrima Mourin, Muhammad Kamruzzaman Mozumder.

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
