## [Decision Letter · Decision Letter 0]

29 Jul 2024

PONE-D-24-24368Development of a New Perceived Injustice Scale for Bangla Speaking PopulationPLOS ONE

Dear Dr. Mozumder,

Thank you for submitting your manuscript to PLOS ONE. After careful consideration, we feel that it has merit but does not fully meet PLOS ONE’s publication criteria as it currently stands. Therefore, we invite you to submit a revised version of the manuscript that addresses the points raised during the review process.

We look forward to receiving your revised manuscript.

Kind regards,

Palash Chandra Banik, MPhil

Academic Editor

PLOS ONE

Journal Requirements:

Reviewers' comments:

Reviewer's Responses to Questions

**Comments to the Author**

1. Is the manuscript technically sound, and do the data support the conclusions?

Reviewer #1: Partly

2. Has the statistical analysis been performed appropriately and rigorously? 

Reviewer #1: Yes

3. Have the authors made all data underlying the findings in their manuscript fully available?

Reviewer #1: Yes

4. Is the manuscript presented in an intelligible fashion and written in standard English?

Reviewer #1: Yes

5. Review Comments to the Author

Reviewer #1: The authors present the development process of a new type of psychological scale and expound on a relatively new psychological concept - perceived injustice. However, the article still requires further modification.

1. In the “background“ section or ”discussion“ section, the author should elaborate on what the specific applied values of "perceived injustice" and the "perceived injustice scale" are in clinical practice and psychological practice. Perceived injustice seems to hold a more significant position in social activities but has less evident roles in clinical applications. The author needs to further argue this point to alleviate the reader’s doubts.

2. The author mentions in the manuscript that there have been previous studies reporting scales related to 'perceived injustice'. Therefore, it is necessary to further elaborate on the similarities and differences between the old and new scales, as well as their respective advantages and disadvantages in practical applications.

3. When the Cronbach's alpha coefficient is excessively high, particularly above 0.95, it may indicate that the items in the scale are overly similar, leading to redundancy. In other words, this suggests that the scale may contain multiple items that measure the same issue, increasing the length of the scale without adding substantial information, thus reducing the efficiency of the assessment. Moreover, an overly high Cronbach's alpha may result from an overestimation of the inter-item correlations, leading to an overly optimistic appraisal of reliability. In the manuscript, the author reports a Cronbach's alpha of 0.971. A deeper analysis of the reasons behind this excessively high coefficient, the accuracy of the data, and considerations regarding whether some items should be removed is warranted.

4. In the “Participants“ section, the author estimated a minimum sample size of 226 participants, however, the actual effective sample size was merely 221, indicating that the sample size did not meet the projected requirement. Furthermore, the manuscript does not mention how many potential participants the researchers approached, how many declined to participate in the study, and how many were excluded and for what reasons.

6. PLOS authors have the option to publish the peer review history of their article (what does this mean?). If published, this will include your full peer review and any attached files.

Reviewer #1: No

---

## [Author Response · Author response to Decision Letter 0]

11 Sep 2024

Response to Reviewer Comments to the Author

Reviewer #1: The authors present the development process of a new type of psychological scale and expound on a relatively new psychological concept - perceived injustice. However, the article still requires further modification.

Thank you for reviewing the manuscript. We have addressed your recommendations which are presented int eh following section. 

1. In the “background“ section or ”discussion“ section, the author should elaborate on what the specific applied values of "perceived injustice" and the "perceived injustice scale" are in clinical practice and psychological practice. Perceived injustice seems to hold a more significant position in social activities but has less evident roles in clinical applications. The author needs to further argue this point to alleviate the reader’s doubts.

Response: Thank you for raising this important point. We were so much involved in the idea that we missed to indicates the applied aspects of the concept. We have added some texts to clarity this.

Perceived injustice seemingly holds a more significant position as a social phenomenon. However, in recent time it is being discussed and studied as a novel and useful concept in medicine and psychiatry (7). (page 4, Introduction)

Moreover, psychotherapeutic interventions (such as Acceptance and Commitment Therapy) are being tested and proven effective in reducing perceived injustice (10). (page 4, Introduction)

With increasing recognition of perceived injustice as a relevant phenomenon in mental health, psychiatry and medicine (4, 7-10), the newly developed tool is expected to be welcomed among the researchers and clinicians from these field of study. (page 15, Discussion).

2. The author mentions in the manuscript that there have been previous studies reporting scales related to 'perceived injustice'. Therefore, it is necessary to further elaborate on the similarities and differences between the old and new scales, as well as their respective advantages and disadvantages in practical applications.

Response: As this is a relatively novel concept, only few tools are available to measure this. A whole section discussed these tools and their context of development as well as prospective usage. To avoid a disproportionately long background section, only key features in a general manner has been discussed. However, in line with the suggestion, we have added the following sentence to further clarify their limitation. 

The tools for measuring perceived injustice as discussed above were developed to fit with specific contexts including chronic pain (11), war and conflict (12), or workplace (13). None of these seems to be a useful tool for assessing perceived injustice in a general context. (page 5) 

3. When the Cronbach's alpha coefficient is excessively high, particularly above 0.95, it may indicate that the items in the scale are overly similar, leading to redundancy. In other words, this suggests that the scale may contain multiple items that measure the same issue, increasing the length of the scale without adding substantial information, thus reducing the efficiency of the assessment. Moreover, an overly high Cronbach's alpha may result from an overestimation of the inter-item correlations, leading to an overly optimistic appraisal of reliability. In the manuscript, the author reports a Cronbach's alpha of 0.971. A deeper analysis of the reasons behind this excessively high coefficient, the accuracy of the data, and considerations regarding whether some items should be removed is warranted.

Response: We are thankful to you for this very important suggestion. As per your suggestion we have further analysed the data and was able to remove a several items. This resulted in a more succinct scale with ten items and we run the reliability and validity analysis again on this 10-item scale. Surprisingly, it not only made the tool more parsimonious, it also resulted in better validity outcome. 

High correlations (r ≥.8) indicate concerns of redundancy of items. Additionally, an initial assessment of internal consistency suggest that the 19-item scale would have an overall Cronbach’s alpha of .971 providing further indication of overly similar items. Therefore, to attain a parsimonious scale, further scrutiny of inter-item correlation matrix was carried out. Correlation coefficient was rounded to single decimal point and multiple items were found with overly high correlation. Item # 10 demonstrates high correlations (r ≥.8) with seven others items (4, 5, 6, 7, 8, 11, 12) which also had high inter-correlation within themselves. Item # 16 shows similar high correlation with two other items (15 & 17). It was obvious that the two key items (10 & 16) would be able to represent the remaining and hence the nine items (4, 5, 6, 7, 8, 11, 12, 15 & 17) was removed. (page 10; Section: Inter-item Correlation) 

For changes in psychometric properties see page 2 (Abstract - Results), page 10 (Corrected Item-total Correlations) page 11 (Factor analysis), page 11-12 (Validity of the Perceived Injustice Scale) and page 12 (Reliability of the perceived injustice scale).

4. In the “Participants“ section, the author estimated a minimum sample size of 226 participants, however, the actual effective sample size was merely 221, indicating that the sample size did not meet the projected requirement. Furthermore, the manuscript does not mention how many potential participants the researchers approached, how many declined to participate in the study, and how many were excluded and for what reasons.

Response: Data were collected from 225 participants (although the target was 226). However, 4 were discarded due to incomplete response. Thus, we fall short of only 5 participants, for which we could not wait any further as this study was done as academic research carried out within strict deadline. Furthermore, our statistical understanding suggests that shortage of 5 participants may not have much of negative impact on the rigor of any analysis conducted in this study. 

Approximately 250 participants were approached, 225 responded.

---

## [Editor Report · Decision Letter 1]

17 Sep 2024

Development of a New Perceived Injustice Scale for Bangla Speaking Population

PONE-D-24-24368R1

Dear Dr. Mozumder,

We’re pleased to inform you that your manuscript has been judged scientifically suitable for publication and will be formally accepted for publication once it meets all outstanding technical requirements.

Kind regards,

Palash Chandra Banik, MPhil

Academic Editor

PLOS ONE
---

## [Editor Report · Acceptance letter]

24 Sep 2024

PONE-D-24-24368R1 

PLOS ONE

Dear Dr. Mozumder, 

I'm pleased to inform you that your manuscript has been deemed suitable for publication in PLOS ONE. Congratulations! Your manuscript is now being handed over to our production team.

Kind regards, 

on behalf of

Dr. Palash Chandra Banik 

Academic Editor

PLOS ONE